# Viewpoint of Chitosan Application in Grapevine for Abiotic Stress/Disease Management towards More Resilient Viticulture Practices

Rupesh Kumar Singh [1,2,*,†], Eliel Ruiz-May [3,†], Vishnu D. Rajput [4], Tatiana Minkina [4], Rosa Luz Gómez-Peraza [5], Krishan K. Verma [6], Mahipal S. Shekhawat [7], Catia Pinto [8], Virgilio Falco [9,10] and Francisco Roberto Quiroz-Figueroa [5,*]

1   InnovPlantProtect Collaborative Laboratory, Department of Protection of Specific Crops, Estrada de Gil Vaz, Apartado 72, 7350-999 Elvas, Portugal
2   Centre of Molecular and Environmental Biology, Department of Biology, Campus of Gualtar, University of Minho, 4710-057 Braga, Portugal
3   Red de Estudios Moleculares Avanzados, Cluster BioMimic®, Instituto de Ecología A. C, Carretera Antigua a Coatepec 351, Congregación el Haya, Xalapa CP 91073, Mexico
4   Academy of Biology and Biotechnology, Southern Federal University, 344090 Rostov-on-Don, Russia
5   Instituto Politécnico Nacional, Centro Interdisciplinario de Investigación para el Desarrollo Integral Regional Unidad Sinaloa (CIIDIR-IPN Unidad Sinaloa), Laboratorio de Fitomejoramiento Molecular, Blvd. Juan de Dios Bátiz Paredes no. 250, Col. San Joachín, Guasave CP 81101, Mexico
6   Key Laboratory of Sugarcane Biotechnology and Genetic Improvement (Guangxi), Ministry of Agriculture and Rural Affairs/Guangxi Key Laboratory of Sugarcane Genetic Improvement/Sugarcane Research Institute, Guangxi Academy of Agricultural Sciences, Nanning 530007, China
7   Biotechnology Unit, Kanchi Mamunivar Government Institute for Postgraduate Studies and Research, Puducherry 605 008, India
8   Associação SFCOLAB—Laboratório Colaborativo para a Inovação Digital na Agricultura, Rua Cândido dos Reis nº 1. Espaço SFCOLAB, 2560-312 Torres Vedras, Portugal
9   Centro de Química de Vila Real (CQ-VR), Universidade de Trás-os-Montes e Alto Douro (UTAD), Quinta de Prados, 5000-801 Vila Real, Portugal
10  Departamento de Agronomia, Universidade de Trás-os-Montes e Alto Douro (UTAD), Quinta de Prados, 5000-801 Vila Real, Portugal
*   Correspondence: rupeshbio702@gmail.com (R.K.S.); fquirozf@hotmail.com or labfitomol@hotmail.com (F.R.Q.-F.)
†   These authors contributed equally to this work.

**Abstract:** Chitosan is a biopolymer with various favorable properties (biotic/abiotic stress mitigation, qualitative improvement, bio-fertilizer, bio-stimulant and postharvest management) to meet multiple agricultural objectives. Grapevine is an important crop and has an enormous impact on the world's economy due to its derived products, notably the different wine styles. In viticulture, chitosan application made significant developments towards higher contents of beneficial metabolites in grape berries as well as stress and postharvest management during recent decades, although the reports are limited. Recent investigations by our group demonstrated chitosan as a potential elicitor molecule at a molecular level and opened the possibility to use chitosan for trunk disease management; moreover, there are not yet any methods to combat trunk diseases in grapevine. The present viewpoint aimed to summarize the different aspects of chitosan application in grapevine in facilitating the development of inclusive and more integrated sanitary viticulture practices in a sustainable manner.

**Keywords:** sustainability; elicitor; secondary metabolites; crop protection

## 1. Introduction

Chitin is an essential component of insect (crustacean) exoskeletons and fungal cell walls, and its deacetylation results into a linear polymer of two sub-units of d-glucosamine and N-acetyl-d-glucosamine that are linked through 1,4-glycosidic bonds, commonly

known as chitosan [1]. It is biodegradable and biocompatible, elicits an antioxidant response, and is economically feasible and environmentally friendly [1–5]. The use of chitosan against plant pathogens was initiated by Allan and Hadwiger [6], and since then, chitosan has become an interesting biomaterial. Plant scientists have been testing this biopolymer to meet various objectives in agriculture for the last few decades [7–10]. Induced defense responses upon chitin and chitosan application led to its development as a bio-fertilizer, bio-fungicide, bio-bactericide, bio-virucide, natural rhizo-bacteria growth promoter, and bioremediation agent [6,11–17], although it has not yet been tested against phylloxera.

The potential use of chitosan in grapevine for crop protection/crop resilience induction is of the utmost importance as grapevine is a major crop across the globe and has a substantial impact on the economy. Winemaking, table grapes, juices, resins and other consumable products are being produced from grape berries, which provide numerous health benefits as grapevines contain diversified secondary metabolites [18]. The International Organisation of Vine and Wine (OIV) listed major grapevine-growing countries (Figure 1) and their primary grape consumption type [19].

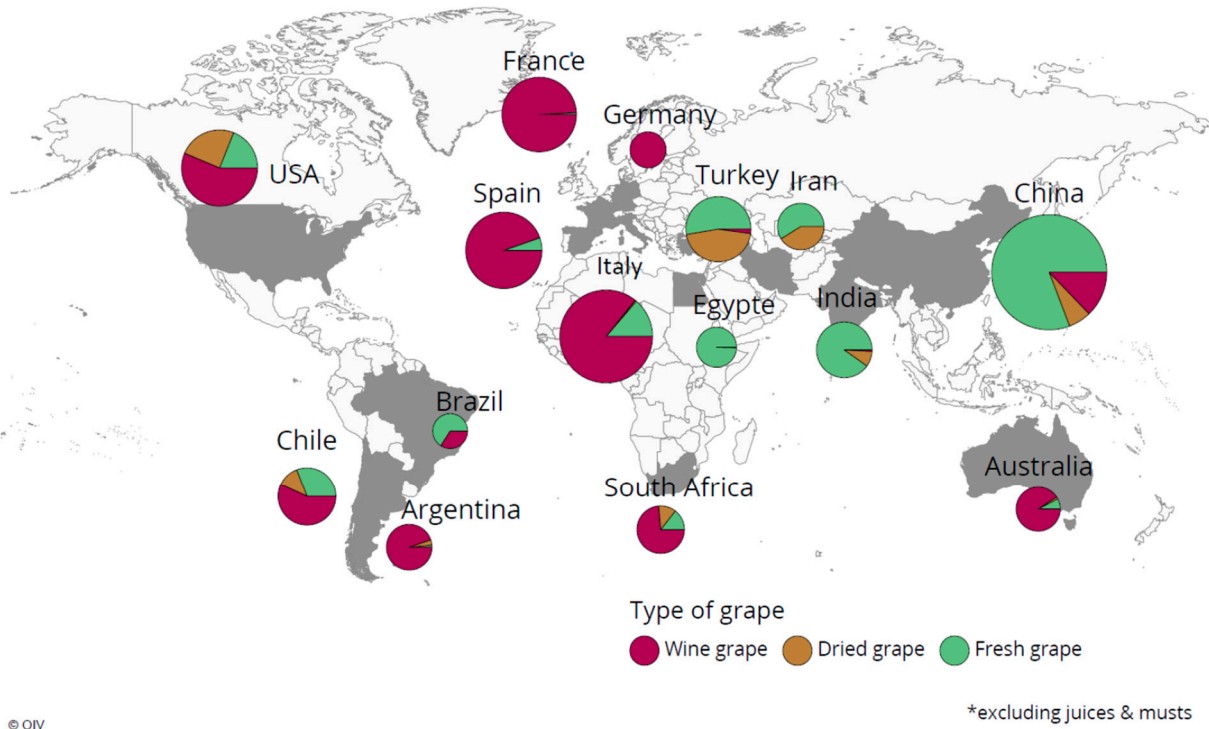

**Figure 1.** Worldwide distribution of grapevine cultivation, adopted from OIV report 2019.

Grapevine is prone to infection from bacteria, viruses, mycoplasma, insects, worms, arthropods, oomycetes and fungi, where fungal and oomycetes pathogens lead to the maximum loss in terms of quality and quantity [20]. Grey mold disease by *Botrytis cinerea* Pers. (teleomorph: *Botryotinia fuckeliana*) (Helotiales: Sclerotiniaceae), a necrotrophic deuteromycete; powdery mildew by *Erysiphe necator*, a biotrophic ascomycete; downy mildew by *Plasmopara viticola* (Peronosporales: Peronosporaceae), a biotrophic oomycete; and trunk diseases by fungal association represent the major diseases worldwide [20]. In general, chemical pesticides and fungicide treatments are needed for fungal disease prevention in grapevine; however, no phytosanitary treatment is available yet for trunk diseases. A few active formulations with different modes of action were found to be effective in avoiding fungal resistance [21,22]. Fungicides and pesticides are potentially hazardous chemicals for winegrower's health and can have serious consequences on environmental health resulting from contaminated water resources and diminished soil microbiota [23–26]. These toxic chemicals may affect the presence of natural yeasts in grapevine, leading to

the deterioration of peculiar aromatic profiles by affecting the grapevine physiology and biochemistry, and subsequently, the wine quality [27–30]. Most importantly, traces of these chemicals identified in wines affect the taste [29].

*Devastating Trunk Diseases in Grapevine*

Esca, Eutypa dieback, and Botryosphaeria dieback are the major trunk diseases that infect mature grapevines, leading to subsequent plant death, and have no cure. Petri disease or black-foot disease caused by species belonging to the Nectriceae family of the fungal order Hypocreales, i.e., *Campylocarpon* Halleen, Schroers & Crous, *Cylindrocladiella* Boesew, *Dactylonectria* Lombard & Crous, *Ilyonectria* Chaverri & Salgado, and *Neonectria* Wollenw. These species infest young vineyards and result into huge economic collapse for vine industries [31]. These grapevine diseases have been the most devastating threat in recent times and are expected to worsen in the near future [32]. Sodium arsenate application was the only applied chemical for trunk diseases; however, its use has been prohibited, resulting in the aggravation of vineyard wood diseases, for example, in Spain, from 1.8% in 2003 to 10.5% in 2007 [32]. Several other factors may also influence the development and spread of these diseases. For example, the introduction of infected vine stocks; cultural practices of improper pruning, leaving the wounds for infection; and exposure to various stress situations to the plant [33]. Trunk disease has arguably been the most destructive type of grapevine disease in the major wine-growing countries for at least the last three decades. The primary reason is that the diseased plants need to be replaced, and such a process accounts for approximately USD 1.5 billion per year [33].

Viticulture scientists worldwide consider the grapevine trunk diseases as a major challenge due to a lack of any precise cure to date. The OIV came forward with a resolution to control the spread of wood diseases in 2006 [34] and in the 2008 and 2011 edition [35]. PROTEC (Vine Protection and Viticulture Techniques) is a very recent draft from OIV to study the effects of the disease worldwide and to develop its mitigation strategies [36].

Altogether, there is an urgent need to develop a sustainable strategy for vineyards to reduce the risks related to human and environmental health. Many methods have been reported for this purpose in the recent past; however, the application of elicitors to induce the plant defense mechanism became the most popular [37–41]. Elicitors are the molecules that induce the secondary metabolite synthesis in plants, thus eliciting the essential pathways towards improved immune responses, and chitosan has been proven to be a potential material for this scenario [42,43]. The present viewpoint aimed to discuss the chitosan applications in grapevine for different objectives and to draw an opinion based on recent investigations by our group with a special focus on devastating trunk diseases.

## 2. Different Objectives for Chitosan Application in Grapevine

### 2.1. Disease Management

The susceptibility of grapevine to various pathogens is well known; moreover, the grey mold disease caused by *Botrytis cinerea* is very common, which frequently affects the different plant parts at different developmental stages [44]. Recently, a commercial acidic solution of chitosan was tested against *B. cinerea* in grapevines, which demonstrated the induced defense responses by upregulating the jasmonic-acid- and ethylene-mediated responses, downregulation of salicylic acid signaling and modulation of trans-resveratrol [45]. The application of chitosan in grapevine as the effective and environmentally friendly approach against *B. cinerea* is suggested.

The elicitation potential of chitosan was assessed in cell cultures of grapevine (*V. vinifera* cv. Limberger) tissues and various parameters were evaluated for, e.g., oxidative damage, extracellular alkalization, and defense responsive gene expression (such as pathogen responsive genes (PR-1, PR-9) and the phenylalanine ammonia-lyase gene (PAL)), suggesting an induced defense responses of chitosan on applied cells [46]. Culture medium supplemented with chitosan-based gel (1.75% *v/v*) was reported to induce the shoot and root biomass and photosynthesis and decrease the development of *B. cinerea* in cultured

plantlets. Moreover, exogenous foliar application of same chitosan gel reduced the disease infestation [47]. Another investigation suggested high protection against grey mold by *B. cinerea* under controlled conditions with chitosan treatment (75–150 mg/l) on grapevine leaves, and a subsequent induction of the activity of key enzymes such as lipoxygenase (LOX), PAL and chitinase markers were observed at a molecular level that eventually inhibited grey mold mycelium development [48]. Chitosan-based application (Biochicol 020 PC) was reported to inhibit the growth of *Phomopsis* (Sacc.) Sacc. *viticola* (Diaporthales: Valsaceae) on stored grapevine shoot cuttings in comparison to the fungicide (azoxystrobin and mancozeb) [49]. Another study suggested a reduced disease level by *B. cinerea* and *P. viticola* on grapevine leaves upon chitosan application [50]. Chitosan induced the PAL enzymatic activity and phytoalexins in grapevine leaves within 2 h and improved the secondary metabolite synthesis and salicylic acid, subsequently enhancing the resistance against fungal pathogen infestation [50]. Different concentrations of chitosan were tested in grapes to evaluate its antifungal potential for *Colletotrichum* sp., and a higher amount of chitosan was observed to reduce the fungal growth significantly [51]. Chitosan-treated suspension cell culture showed increased expression of PR-10 family transcripts [52]. Treating fruits on the field with chitosan could directly inhibit the growth of *B. cinerea* and could be an alternative to traditional fungicides in post-harvest disease control of grapes [53]. Vineyards of Chardonnay and Sauvignon blanc grapevines were treated with chitosan (at concentrations greater than 0·125 g/L), and the results suggested a reduced infestation of *B. cinerea* in grape bunches [54]. Iriti el al. [55] tested a new chitosan formulation (Kendal Cops®, Kc) in grapevine and demonstrated higher resistance against powdery mildew infestation. *Sphaceloma ampelinum* de Bary (Myriangiales: Elsinoaceae) infection was reduced significantly, i.e., by 75%, upon chitosan treatment, eventually reducing the anthracnose grapevine disease by increasing the synthesis of salicylic acid [56]. Bois noir infected vineyards (cv. Chardonnay) were treated with chitosan and exhibited an improvement towards disease control [57]. Suspension cell culture of *V. vinifera* showed higher expression of chitinase and β-1,3-glucanase upon chitosan treatment; these two proteins are well-known for pathogen response [58]. Moreover, chitosan application significantly reduced the grapevine downy mildew upon application in field [59]. Recently, the study of proteomics and metabolomics was performed in pot-grown plants (*V. vinifera* L. cv. Ortrugo, grafted on 420A rootstock) treated with 0.03% chitosan solution. Metabolomics analysis displayed the overproduction of triterpenoids, betulin, erythrodiol, uvaol, and oleanolate. These compounds are known to play antifungal, antimicrobial, and insecticidal roles. In the same study, the proteomic profile revealed higher superoxide dismutase (SOD) and PAL activity in chitosan-treated grapevines, thus inducing the reactive oxygen species (ROS) and PAL pathway [60]. Lucini et al. [61] reported an increased defense response by elicited accumulation of phytoalexins in chitosan-treated grape bunches. In addition, differential oxidative balance in reference to the control was observed, and it was suggested that chitosan modulated the oxidative stress responsive enzymes and increased the accumulation of phenylpropanoid and triterpenoids phytoalexins.

### 2.2. Induced Secondary Metabolism and Stress Management

The induction of secondary metabolite synthesis has shown the potential for qualitative and quantitative improvements in viticulture practices [62]. Chitosan was tested on grapevine leaves as an exogenous application, and the findings suggested a higher accumulation of trans and cis-resveratrol, viniferin and piceid [62]. Chitosan pre-harvest treatment improved the quality attributes in table grapes [63]. Chitosan was tested in in-vitro grown grapevine cuttings and resulted in the reduction of drought and temperature stress. Moreover, a higher root growth and plant development was recorded in chitosan-treated cuttings [64]. Physiological improvements were observed in pre-harvest chitosan-treated grapes [63]. Different components of grapes were studied for their antioxidant potential upon chitosan application [65]. Chitosan induced the stilbenes accumulation in suspension cell cultures of *V. vinifera* cv. Barbera grape [52]. Suspension cell cultures and

regenerated calli (*V. vinifera* cv Italia) were tested for chitosan elicitation towards stilbene synthesis. However, the increased amount was not significantly different in comparison to control [66]. Higher antioxidant activity was observed upon the application of a new chitosan formulation product [55]. Maura Ferri et al. [67] reported induced mono-glucosylated stilbenes in grape cells in a bed batch bioreactor. Chitosan was used as an activator in comparison to conventional fungicides, and the findings suggested improved sterol levels (especially *β*-sitosterol) in grapes and derived micro-vinifications [68]. Chitosan induced the amount of total acetals, alcohol and sensory profile upon application in vineyard cultivating *V. vinifera* cv. Groppello Gentile [69]. 2D-PAGE coupled to mass spectrometry analysis showed that chitosan-treated (50 μg/mL) cell cultures (*V. vinifera* cv. Barbera) exhibited three soluble stilbene synthase protein spots, four stilbene synthase spots, and four spots of ATPase membrane subunits in the protein profiling when compared to the control. The study demonstrated that proteomic alterations may improve the grape berries qualitative characters including taste, flavor, organoleptic, and nutraceutical metabolomics contents [70]. Cell suspension cultures treated with chitosan resulted in the overaccumulation of stilbene and trans resveratrol, in the cells and culture medium, respectively, while the biomass of cell cultures did not change by treatment with chitosan. Organic vineyards containing Sangiovese and Cabernet Sauvignon red grape cultivars were sprayed on the canopy with chitosan, and the observations suggested significant differences in the content of (+)-catechin, (−)-epicatechin and procyanidin B2 in grapes berries [71]. Another chitosan application on grapevine showed higher accumulation of secondary metabolites in grapes and derived wines [72]. Canopy treatment in organic vineyards (*V. vinifera* cv. Cabernet Sauvignon) showed a significant increase in γ-aminobutyric acid (GABA), amino acids, amines, phenolic acids, and nitrogenous compounds, suggesting an improved response towards stress conditions in grapevine [71]. Thompson seedless grapevines were tested with chitosan and the agronomical parameters were observed to be significantly improved in the two-year study [73].

Higher terpenoid (ursolate, oleanoate and betulinate) levels were recorded in grape bunches upon chitosan treatment [61]. Volatile compounds were recorded in a higher concentration upon chitosan application in Tempranillo grapes [74]. The total phenolic content and total tannins were recorded as over-accumulated in the two Portuguese grapevine varieties [75]. Overall, the results demonstrated that chitosan has a stimulatory effect on the accumulation of phenolic compounds, including anthocyanins, which is mediated by the modifications in the transcription of key genes involved in their biosynthesis and transport in grape berries [75,76]. Furthermore, the antioxidant potential was improved in different tissues of *V. vinifera* cv. Touriga Franca and *V. vinifera* cv. Tinto Cao [75]. There were increased antioxidant and antimicrobial activities in different grape components treated with chitosan solution [77], and the seed extracts showed the highest antioxidant and antimicrobial activities. The studied individual components obtained from chitosan-treated grapevines could represent added value due to an increased antioxidant and antibacterial potential. The phenolic compounds found in the different components may be used in food and pharmaceutical industries as natural food preservers and antibiotic adjuvants [77].

## 3. Postharvest Management

Postharvest management in grapes is very important in order to maintain quality between harvesting and consumption. Grey mold is a major cause of the decay of perishable fruits such as grapes, in the field as well as during storage and transportation. Chitosan pre-harvest treatment was applied to evaluate the antifungal potential in table grapes, and the findings suggested a lower infestation of grey mold [78]. Postharvest coating with chitosan on grape bunches reduced the rate of fruit decay [63]. Chitosan formulation effectively reduced the gray mold disease in stored grapes, decreased $CO_2$ and oxygen exchange, and prevented lesions in grape berries [79,80]. Coating with chitosan formulation resulted in a reduction in fruit decay and subsequent investments in quality management in grapes during storage, and such an approach may be a promising

postharvest management method [81]. Chitosan coating was applied on grape berries (*V. vinifera* L. cv. Heiti) during storage at room temperature, and the observations suggested a lower respiratory metabolism and water loss, decreased titratable acid content, lower malonaldehyde accumulation, and improved SOD activity. Therefore, the treatment was found to have potential in the postharvest quality management of grape berries [82]. Gray mold is a major cause for the decay of perishable fruits such as grapes, in the field as well as during storage. Chitosan coating was found to be effective against the reduction in the gray mold lesions and for the induction of resistance in grape berries; hence, it can be considered as a potential protection agent for postharvest management in the grape berry industry [80]. Bunches of 'Thompson Seedless' grape berries were applied with three chitosan products (OII-YS, Chito Plant, and Armour-Zen) at berry set, pre-bunch closure, veraison, and at two or three weeks before final harvesting, and stored at 2 °C for 6 weeks. Chitosan application reduced the postharvest gray mold, improved the endochitinase activity, decreased the hydrogen peroxide content, and increased the quercetin, myrcetin, and resveratrol contents in the berries [83]. Chitosan nanoparticle formulation was used to make a coating on grape berries stored at 12 °C and 25 °C, and the potential for postharvest life, and total soluble solids, pH, titratable acidity, reducing sugar content, moisture content, and sensory characteristics of grapes were studied. The results demonstrated that the edible chitosan nanoparticle coating delayed the ripening of grapes and sensory characteristics, thus maintaining the quality parameters during storage [84].

## 4. The Need for Sustainable Methods to Replace the Chemicals Used in Vineyards

Insecticides, miticides, herbicides, pesticides, and fungicides are toxic chemicals for human and environmental health and have been used extensively in grapevines worldwide [85,86]. Approximately 10 applications are essential in vineyards every year to prevent the disease infestations. These toxic chemicals have a drastic impact on the health of wine growers, as the direct exposure and contamination of surface water possess a potential threat to the environmental health. Moreover, traces of these chemicals have been found in different parts of grape plants, including berries [87]. Accumulative risks of these substances have provoked researchers to find unique sustainable solutions for multiple disease- and pest-related problems in vineyards [88]. Grape growers need an effective set of tools to reduce the disease in vineyards as well to eliminate chemical application-related health hazards. Some barrier methods have been proposed earlier, such as the use of plastic nets and covers to deter insects; however, such measures are not practically feasible on a large scale. Breeding traditional wines with resistant ones may provide an alternative, but this can also spoil the flavor of traditional wines. Genetic engineering-based approaches were also considered to promote the disease resistant traits; however, strong laws against GM crops and the societal acceptance of it represent major hindrances in this direction. Biofungicides and biopesticides were reported to be the best alternates to overcome this issue [89,90], but there were no significant answers for the control of downy mildew (*P. viticola*) in grapevines. Biocontrol agents have limitations due to their cost, quick degradability in fields, and temperature variations. Therefore, a significant potential biomaterial needs to be identified to overcome the multiple problems.

## 5. Future Prospects: An Opinion with Special Focus to Trunk Diseases Management

Keeping the above-discussed facts in mind, chitosan may be a potential biopolymer and act as a base material in developing new plant protection product for grapevines. It has shown positive effects towards various grape diseases upon its application, although the molecular mechanism behind this action is still being explored. A very recent report by our group showed the effects of chitosan application in *V. vinifera* cv. Touriga Franca and *V. vinifera* cv. Tinto Cao in field conditions [75].

The results from this study suggested an increase in the total phenolics, total tannins, and total anthocyanins in the different tissues (Figure 2). This improves the quality parameters of grapes and derived wines, as well as promoting the resilience in the plants. The

antioxidant potential was also recorded as higher upon chitosan treatment as a modulation of genes belonging to ROS pathway was noticed. In particular, Cu-Zn/SOD, catalase, amine oxidase, peroxidase (POD) and polyphenol oxidase (PPO) were observed as up-regulated in the trunks in both of the varieties upon chitosan treatment. The total tannins were observed to be much higher in trunks of chitosan-treated plants in comparison to control. Such a plant immune response suggests an improved trunk health of grapevines; however, a similar possible solution in terms of preventing grapevine trunk diseases is yet to be explored. Another study by our group demonstrated the upregulation of basic genes of phenolic compounds during the *veraison* stage in the Tinto Cao variety [76].

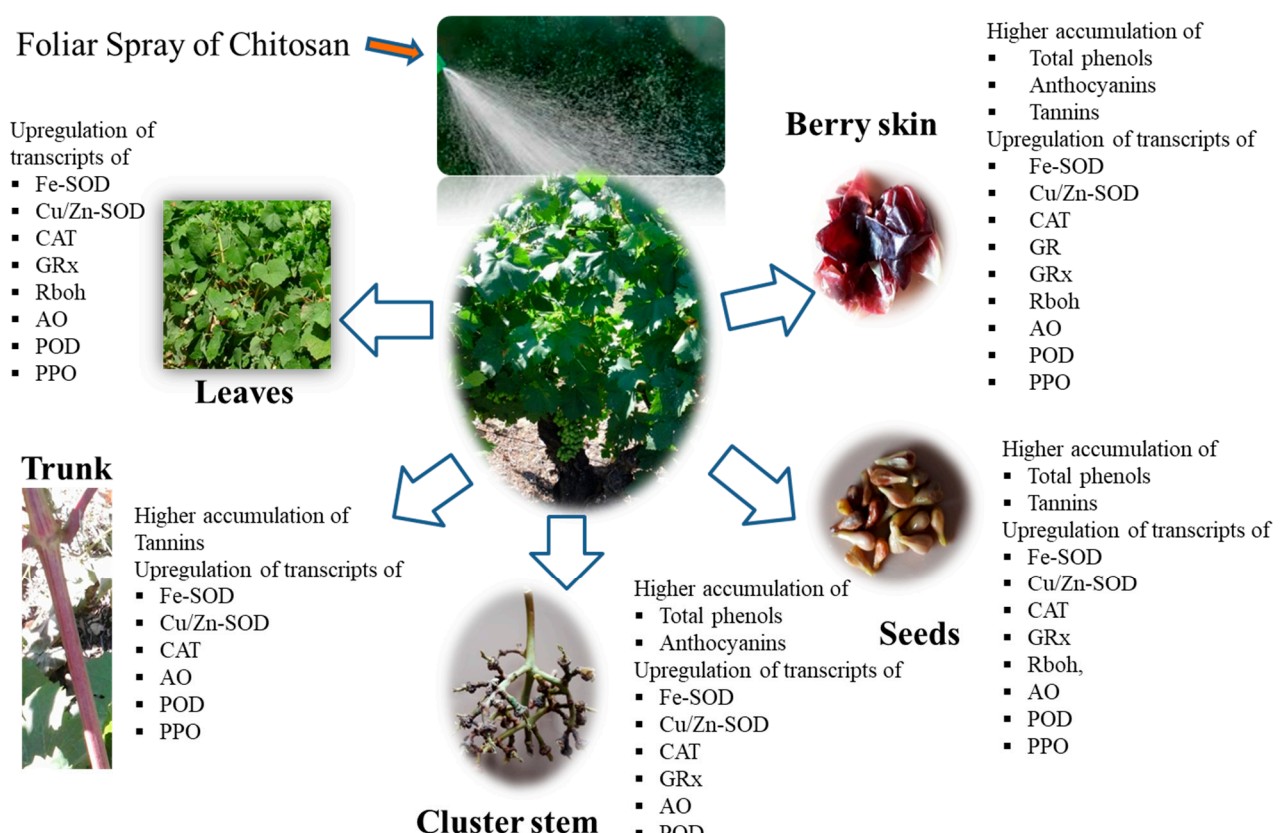

**Figure 2.** Diagrammatic representation of effect of chitosan application in grapevine in tissue specific manner at a molecular and biochemical level where different genes denominated as iron-superoxide dismutase (Fe-SOD), copper-zinc-superoxide dismutase (Cu/Zn-SOD), catalase (CAT), glutathione reductase (GR), glutaredoxin (Grx), respiratory burst oxidase (Rboh), amine oxidase (AO), peroxidase (POD) and polyphenol oxidase (PPO).

Increased antimicrobial activity was also reported by chitosan application in different grape components of the Sousao grape variety. Field studies by our recent investigations suggested that chitosan can be used to develop new plant protection products alone or with some formulations to control the diseases in a sustainable manner while minimizing environmental and human health hazards. For example, copper is widely used in grapevines for disease control while the application is limited by European regulation and thus generates an urgent need to find an alternative method.

Chitosan-based nanopesticides could be a promising approach in controlling trunk diseases in grapevines due to the antimicrobial and plant-growth-promoting properties of nanoparticles (NPs) [91–94]. One of the most popular biomaterials in nanotechnology is chitosan, as it is a non-toxic to human beings. Chitosan-nanocomposites could be integrated as biocides, disease controllers, and/or plant growth promoters [95].

Several researchers have synthesized chitosan-based nanocomposites which showed antifungal properties and effectiveness to control the plant pathogenic fungal diseases [93]. Phytopathogens *Eutypa* Fries (Xylariales: Diatrypaceae) and *Botryosphaeria* Ces. & De Not. (Botryosphaeriales: Botryosphaeriaceae) and those causing Esca disease are the major fungal diseases in grapevines. More than 30% of all cultivated plant diseases are infected by pathogenic fungi. Earlier reports demonstrated that Cu-NP-based chitosan composite controlled the *Fusarium oxysporum* (Link) Schlechtendal (Hypocreales: Nectriaceae) disease by 61.1% in tomato [92,96]. In vitro studies indicated an approximately 80% reduction in *Fusarium circinatum* Nirenberg & O'Donnell and *Diplodia pinea* (Desm.) Kickx (Lecanorales: Acarosporaceae) fungal mycelial growth by chitosan and Ag-'NP mixtures in solution application [91]. The solution of oleoyle (O)-chitosan NPs showed strong antifungal activity (78.16–79.10%) on mycelium growth of *Botryosphaeria dothidea* Ces. & De Not [97]. The combination of Cu-chitosan NPs was also effective against *Curvularia* Boedijn (Pleosporales: Pleosporaceae) leaf spot of maize when applied as a foliar application [98].

Chitosan-based nanocomposites could be used as safer bio-pesticides to replace synthetic pesticides due to their biodegradability and biocompatibility properties. The available experimental data that has been reviewed in this work proved it as an effective bio-pesticide as well as an elicitor of plant antioxidative response defense mechanism when compared to the bulk material/individual application. Nevertheless, the use of chitosan-based nanocomposite could be a promising replacement to synthetic agrochemicals in grapevine disease management. However, much more data are needed to understand the functionality of these composites to scale-up synthesis process and customize stable and desired formulations.

## 6. Conclusions

In conclusion, chitosan may be an excellent biomaterial for future crop protection method developments for winegrowers worldwide. Chitosan exhibits potential in developing a strategic pathway, and it shows promise as a unique measure to answer multiple problems in grapevines, especially for the management of grapevine trunk diseases. The exact molecular mechanism behind chitosan–host plant interaction is still not very clear, and an inclusive transcriptomic, metabolomics, and proteomic approach may unveil the basic molecular mechanism for further facilitating the development of its application methods.

**Author Contributions:** R.K.S., E.R.-M. and F.R.Q.-F. conceptualize and wrote the manuscript. V.D.R., T.M., R.L.G.-P., K.K.V., M.S.S., C.P. and V.F. reviewed the manuscript thoroughly. All authors have read and agreed to the published version of the manuscript.

**Funding:** V.D.R. and T.M. acknowledge support by the laboratory «Soil Health» of the Southern Federal University with the financial support of the Ministry of Science and Higher Education of the Russian Federation, agreement No. 075-15-2022-1122.

**Acknowledgments:** KitoZyme (Belgium) is gratefully acknowledged to provide the chitosan material.

**Conflicts of Interest:** The authors declare no conflict among them for the publication of the present opinion paper.

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
