# Peer review of "Viewpoint of Chitosan Application in Grapevine for Abiotic Stress/Disease Management towards More Resilient Viticulture Practices"

_agriculture, doi:10.3390/agriculture12091369_

Round 1
Reviewer 1 Report
Manuscript deals with a very current topic from the point of view of viticultural research. And includes some authors' previous research on chitosan application.
Below are some minor points to be considered before publication:
Line 63-80 I suggest mentioning phylloxera
Line 196-197 Is stilbene synthase a repetition here?
Line 248, 251, 252, 258 add a space between grape and fruits (all together is another type of fruit)
Author Response
Dear reviewer-1
We thank you for the valuable suggestions and we tried to address all of them, please find the detail description here below.
Comment 1- Line 63-80 I suggest mentioning phylloxera
Author´s response- Thank you for this comment, we have mentioned phylloxera, and although there is no report regarding chitosan mediated control of phylloxera. Moreover it has been a nice idea for our future studies to test some chitosan formulations in the soil, to test against phylloxera management.
Comment 2- Line 196-197 Is stilbene synthase a repetition here?
Author´s response- All the repetitions have been checked and changed accordingly.
Comment 3- Line 248, 251, 252, 258 add a space between grape and fruits (all together is another type of fruit).
Author´s response- Thank you for this comment, all the “grapefruits” were changed to grape berries.
On behalf of all the authors, I thanks again and appreciate your consideration.
With best regards
Reviewer 2 Report
The manuscript addresses a topic of importance for viticulture, with special emphasis on sustainable management techniques through the use of chitosan. The text is well written, but there are some points that need improvement, which are detailed below.
1.- Verify email format
2.- Keywords: Change the words "Chitosan" and "Viticulture" because they are already in the title.
3.- Line 58-59, mention the acronym OIV.
4.- Line 91, there is an extra space.
5.- Line 100-101, Delete "International organization of Vine and Wine" and name it only with the acronym OIV.
6.- Line 144, see format of "Colletotrichum".
7.- Line 150, verificar el punto en "0.125".
8.- Line 172, see format of "phenylpropanoid"
9.- Line 199, check the phrase "... compared with the to the control".
10.- Line 248-249, repeated phrase, already mentioned in lines 235-236. Please verify.
11.- Line 173, Regarding abiotic stress, please mention examples of abiotic stress where what they mention could be useful, thermal stress, frost, water stress, specify.
12. Line 264, About the title, isn't chitosan a chemical? Modify the title, refer to products of chemical synthesis.
13. Line 348 (conclusions), Please conclude on what is developed in the writing, without directly mentioning the commercial name of the products used in the tests carried out and already published. The way the conclusion is written can lead the reader to believe that an "advertisement" is being made about a commercial product, losing the objectivity of the work. In this aspect, it is important to mention in conclusions future perspectives, lines to work on, with special emphasis on the sustainability of the vineyards.
Author Response
Dear reviewer-2
We thank you for the valuable suggestions regarding the manuscript “Viewpoint of chitosan application in grapevine for abiotic stress/disease management towards a more resilient viticulture practices”. We tried our best to address all of them, please find the detail description here below.
Comment 1.- Verify email format
Author’s response- All emails were verified, thank you.
Comment 2.- Keywords: Change the words "Chitosan" and "Viticulture" because they are already in the title.
Author’s response- The words "Chitosan" and "Viticulture" has been deleted from keywords, thank you.
Comment 3.- Line 58-59, mention the acronym OIV.
Author’s response- Has been mentioned accordingly, thank you.
Comment 4.- Line 91, there is an extra space.
Author’s response- Has been formatted.
Comment 5.- Line 100-101, Delete "International organization of Vine and Wine" and name it only with the acronym OIV.
Author’s response- Has been changed accordingly, thank you
Comment 6.- Line 144, see format of "Colletotrichum".
Author’s response- Has been confirmed, thank you
Comment 7.- Line 150, verificar el punto en "0.125".
Author’s response- Verified and edited accordingly, thank you.
Comment 8.- Line 172, see format of "phenylpropanoid"
Author’s response- Verified and edited accordingly, thank you.
Comment 9.- Line 199, check the phrase "... compared with the to the control".
Author’s response- Verified and edited accordingly, thank you.
Comment 10.- Line 248-249, repeated phrase, already mentioned in lines 235-236. Please verify.
Author’s response- The repetition has been deleted accordingly, thank you.
Comment 11.- Line 173, Regarding abiotic stress, please mention examples of abiotic stress where what they mention could be useful, thermal stress, frost, water stress, specify.
Author’s response- Has been changed to specific stress conditions accordingly, thank you.
Comment 12. Line 264, About the title, isn't chitosan a chemical? Modify the title, refer to products of chemical synthesis.
Author’s response- Chitosan is a biopolymer which has been extracted from different organism. Its non-chemical property and abundance in nature made it very popular material to use it towards sustainable agriculture, thank you.
Comment 13. Line 348 (conclusions), Please conclude on what is developed in the writing, without directly mentioning the commercial name of the products used in the tests carried out and already published. The way the conclusion is written can lead the reader to believe that an "advertisement" is being made about a commercial product, losing the objectivity of the work. In this aspect, it is important to mention in conclusions future perspectives, lines to work on, with special emphasis on the sustainability of the vineyards.
Author’s response- The conclusion has been modified towards the objectives and information regarding chitosan provider has been deleted in order to keep it focused.
Thank you
